# Analysis of Inbreeding Effects on Survival at Birth of Pannon White Rabbits Using the Inbreeding-Purging Model

György Kövér [1], Ino Curik [2], Lubos Vostry [3], János Farkas [1], Dávid Mezőszentgyörgyi [1] and István Nagy [1,*]

[1] Department of Animal Science, Hungarian University of Agricultural and Life Sciences, Kaposvar Campus, Guba S. 40, 7400 Kaposvar, Hungary
[2] Faculty of Agriculture, University of Zagreb, Svetošimunska Cesta 25, 10000 Zagreb, Croatia
[3] Faculty of Agrobiology, Food and Natural Resources, Czech University of Life Sciences, Kamycka 129, 16500 Prague, Czech Republic
[*] Correspondence: nagy.istvan.prof@uni-mate.hu

**Abstract:** Mating between related animals is an inevitable consequence of a closed population structure especially when it coincides with a small population size. As a result, inbreeding depression may be encountered especially when considering fitness traits. However, under certain circumstances, the joint effects of inbreeding and selection may at least partly purge the detrimental genes from the population. In the course of this study, our objective was to determine the status of purging and to quantify the magnitude of the eliminated genetic load for the survival at birth of Pannon White rabbit kits maintained in a closed nucleus population. The evolution of the survival at birth was evaluated by applying the PurgeR R package based on the inbreeding-purging model. In the period from 1992 to 2017, 22.718 kindling records were analyzed. According to the heuristic approach, the purging coefficient reached the maximum possible value of 0.5 when estimating between 1992 and 1997. Based on the expected fitness over generations and on the expressed opportunity of purging, the beneficial effects of purging could be expected after 10 generations. The proportion of the purged genetic load could be between 20% and 60%. While the results obtained are not entirely conclusive, they do raise the possibility that some of the inbreeding load was caused, at least in part, by genes that could be successfully removed from the population by purging.

**Keywords:** inbreeding-purging model; purging coefficient; expressed opportunity of purging

## 1. Introduction

Inbreeding is caused by the mating of related animals which is an inevitable consequence of limited diversity in a closed population. The result of mating related animals is that the offspring from inbred mating presents a higher degree of autozygosity across the genome, which results in unmasking recessive deleterious mutations or losing the advantage of alleles with heterozygous superiority [1,2]. The most common consequence of inbreeding is that inbred progenies are often affected by inbreeding depression, which results in a reduction in the phenotypic yield of fitness-related traits [3,4]. Inbreeding and inbreeding depression have been studied extensively for more than a century, and even Darwin (1883, 1892) [5,6] was one of the first to experimentally study the effects of inbreeding, especially in plants. Nevertheless, inbreeding depression is a ubiquitous phenomenon that is also observed in animal populations. The main results of studies on inbreeding depression in captive (e.g., zoo) and wild populations based on both pedigree and genomic analyses have been summarized in reviews [7–10]. Evidence on inbreeding depression in livestock populations is also abundant, with good reviews in the literature [11,12] and a comprehensive meta-analysis including 154 studies published between 1990 and 2020 [13].

However, the fitness decline with increasing inbreeding can be reduced by purging (i.e., purifying selection facilitated by inbreeding). Purging can be effective when the average effect of deleterious mutations is strong (relative to the effective population size);

inbreeding occurs gradually, over several generations, and the population is sufficiently isolated so that purged deleterious alleles are not reintroduced by immigration [14]. While the latter evidence is controversial [15], the beneficial effects of purging were first reported in a small captive population of Speke's gazelle [16,17] where the reproductive performance of the population was improved within a few generations. According to Templeton and Read [16,17], selection and inbreeding were combined to eliminate the deleterious alleles. The consequences of genetic purging on captive breeding and restoration programs were reported by Leberg and Firmin [18] and by Pérez-Pereira [19]. Although purging has extensively been analyzed both in captive and wild populations [20,21], its occurrence could mostly be detected in laboratory populations [22,23]. On the contrary, in domesticated species, indication of purging is rare [24–26]. With regard to the detection of purging, the so-called "inbreeding-purging" method was recently proposed by García-Dorado [27] and the favorable characteristics of this method compared to former procedures based on ancestral inbreeding [14,20] were challenged by López-Cortegano et al. [28].

The study by Curik et al. [24] was based on ancestral inbreeding, and given the recent developments mentioned above, our goal was to re-analyze the dataset [24] using the inbreeding-purging model to complement and improve our earlier findings on purging. The most important objective of this study was to quantify inbreeding depression with special attention to the possible detection of the effect of purging on the survival rate at birth, determining genetic background using a statistical method which—according to our knowledge—has not yet been applied in animal science.

## 2. Materials and Methods

### 2.1. Population and Data Information

The breeding of the Pannon White rabbit population started at the Kaposvár University in the late 1980s and it was officially registered as a Hungarian rabbit breed in 1992. It has been selected as a closed population ever since. In order to keep the inbreeding rate low, a circular mating scheme was used where at the time of population foundation it was sorted into four groups. Bucks are mated to does of the adjacent group: male_group$_1$ × female_group$_2$; male_group$_2$ × female_group$_3$; male_group$_3$ × female_group$_4$; male_group$_4$ × female_group$_1$. All progeny receives the group number of the buck.

Since it was established, the breeding objectives of the breed have changed several times. Since 2010, the selection traits have been litter weight (measured at day 21 after kindling) and thigh muscle volume (based on in vivo computer tomography measurements) [29]. As mentioned in the introduction section, the analyzed data and pedigree was identical to that of the study of Curik et al. [24] Thus, 22.718 kindling records collected between 1992 and 2017 were analyzed. These kindling records were multiplied according to the number of the total kits born. The resulting "individual" kindling dataset contained 203,065 records. All records obtained originated with 1421 bucks and 5339 does. The pedigree was extended with artificial litter identity codes in order that litter inbreeding could also be taken into account; there were 29,802 records altogether.

### 2.2. Inbreeding-Purging Analysis

The analyzed fitness trait was survival of the rabbit kits at birth (W), which is a binomial variable. The survival of the rabbit kits at birth was predicted using a formula adapted from García-Dorado et al. [30]:

$$W = W_0 e^{-(\delta g + \delta_M g_M + S_2 Season + P_2 Parity2 + P_3 Parity3 + P_4 Parity4)}$$

where $\delta$ and $\delta_M$ are inbreeding loads ascribed to the effects of the deleterious alleles in the genotype of the individuals' and that of their dam; g is the purged inbreeding coefficient representing the conventional Wright inbreeding coefficient (F) adjusted by the expected frequency of the purged deleterious alleles relative to their value before purging. The

adjustment is made according to the purging coefficient d, which produces the best fit to the observed consequences of purging on fitness.

- The value of the purging coefficient (d) was estimated using a heuristic approach suggested by García-Dorado et al. [30] covering the interval 0–0.5.
- For each d value assumed, this approach computes the gi value of each individual, where $g_i$ was calculated by the ip_g() function of the purgeR R package.
- In our case, the fitness trait (W) is the survival. Probit regression (glm function, stats package of R) was used to find suitable initial values for the non-linear regression model.
- The non-linear regression method was used to find the more accurate values of the coefficients of the model (nls function, stats package of R).
- The corrected Akaike information criterion AICc value was calculated by the AICcmodavg package.

The estimated value of the purging coefficient (d) was determined by the minimum of the AICc values. Additionally, d > 0 was tested by applying the $Chi^2$ statistic [31].

With regard to the season of kindling (summer or otherwise) and parity of the rabbit doe (merged to four groups), a dummy variable (zero and one representing $Season_1$ and $Season_2$, respectively) and a set of three dummy variables were used to represent four level parity.

The model with the minimum AICc value also provided the values of coefficients ($W_0$, $\delta$, $\delta_M$, $S_2$, $P_2$, $P_3$, $P_4$).

The effective population size (Ne) was determined by the pop_Ne() function of the PurgeR R package. These values were used for fitness prediction. Based on the fitted model, the estimated $W$ values were calculated. The $W'_0$ value is the average (and its standard error) of the W values for the non-inbred animals (f = 0 and $f_M$ = 0).

In addition, based on the pedigree, the expressed opportunity of purging ($O_{Ei}$) was also determined.

$$O_{Ei} = \sum_j 2F_{i(j)}F_j$$

where $F_{i(j)}$ is the probability of an allele in $i$ being derived from an allele in $j$ and being autozygous in $i$, which provides an estimate of the expected reduction of the inbreeding load using the function io_op().

A complementary analysis was also performed, determining the proportional reduction of the inbreeding load as $g_t/F_t(1 - F_t)$ [27] with the proposed values of d = 0 and d = 0.5.

The inbreeding-purging model was applied to analyze the survival rate of rabbit kits at birth in two time periods: one covering the period from 1992 to 1997 and the other from 1992 to 2017. This was done because in our previous study [24], purging was detected only in the first period (from 1992 to 1997). All these analyses were performed using the PurgeR R package [32].

## 3. Results

The effects of the examined factors on the survival of the rabbit kits at birth in the first and in the whole period are given in the Table 1. Based on the heuristic approach [30], the purging coefficients in the first (1992–1997) and in the whole (1992–2017) period coinciding with the lowest AICc values gave d = 0.5 and 0.0, respectively. Nevertheless, it has to be noted that the statistical test (examining if d = 0) was not significant in either period. It can also be seen that the maternal inbreeding did not affect the rabbit kits' survival in either period. The lowest AICc values were received in both periods when the model contained individual inbreeding, season of kindling and parity number of rabbit does, all three of which significantly affected the survival rate of kits at birth. As to the individual inbreeding effect, there was a large depression in the first period but a positive inbreeding effect on the fitness trait was detected when examining the whole period (1992–2017) (Table 1).

**Table 1.** Inbreeding-purging parameters estimated for survival of kits at birth in two analyzed periods.

| d (P) | δ (Se) | $\delta_M$ (Se) | $S_2$ (Se) | $P_2$ (Se) | $P_3$ (Se) | $P_4$ (Se) | $W_0$ (Se) | $W'_0$ (Se) | AICc |
|---|---|---|---|---|---|---|---|---|---|
| Analyzed pedigree for the period between 1992 and 1997 | | | | | | | | | |
| 0.500 (0.294) | **0.466** (0.125) | NA | NA | NA | NA | NA | **0.945** (0.001) | 0.945 (0.000) | −4535.21 |
| 0.500 (0.294) | **0.466** (0.125) | 0.002 (0.169) | NA | NA | NA | NA | **0.945** (0.001) | 0.945 (0.000) | −4533.21 |
| 0.5000 (0.294) | **0.445** (0.125) | 0.003 (0.169) | **0.013** (0.003) | NA | NA | NA | **0.949** (0.001) | 0.945 ($2.76 \times 10^{-5}$) | −4553.38 |
| 0.5000 (0.289) | **0.442** (0.125) | −0.054 (0.169) | **0.012** (0.003) | −0.015 (0.004) | −0.024 (0.003) | −0.033 (0.004) | **0.931** (0.003) | 0.946 ($6.03 \times 10^{-5}$) | −4630.67 |
| 0.5000 (0.290) | **0.442** (0.125) | NA | **0.012** (0.003) | −0.015 (0.004) | −0.024 (0.003) | −0.032 (0.004) | **0.931** (0.003) | 0.946 ($6.03 \times 10^{-5}$) | −4632.57 |
| Analyzed pedigree for the period between 1992 and 2017 | | | | | | | | | |
| 0.2404 (0.158) | **−0.093** (0.016) | NA | NA | NA | NA | NA | **0.946** (0.001) | 0.946 (0.000) | −38,718.32 |
| 0.1019 0.289 | **−0.101** (0.023) | 0.027 (0.024) | NA | NA | NA | NA | **0.946** (0.001) | 0.946 (0.000) | −38,717.25 |
| 0.0000 | **−0.098** (0.023) | 0.028 (0.023) | **0.015** (0.001) | NA | NA | NA | **0.950** (0.001) | 0.946 ($2.92 \times 10^{-5}$) | −38,866.47 |
| 0.0000 | **−0.085** (0.023) | 0.018 (0.024) | **0.015** (0.001) | −0.018 (0.002) | −0.017 (0.001) | −0.014 (0.001) | **0.937** (0.001) | 0.946 ($4.33 \times 10^{-5}$) | −39,012.55 |
| 0.0000 | **−0.071** (0.013) | NA | **0.015** (0.001) | −0.018 (0.002) | −0.017 (0.001) | −0.014 (0.002) | **0.937** (0.001) | 0.946 ($4.34 \times 10^{-5}$) | −39,013.98 |

Significant estimates are bold while all other values are not significant ($p > 0.05$); d is the purging coefficient; P is the observed level of significance of the test if d = 0; δ is the rate of litter inbreeding depression; $\delta_M$ is the rate of dam inbreeding depression; $S_2$ is the season effect compared to summer; $P_2$, $P_3$ and $P_4$ are parity effects compared to the first parity; NA indicates where estimates were not applicable. The observed level of significance (P) and the standard error (Se) of the inbreeding-purging parameters are given in brackets.

Because of the characteristics of the applied equation characterizing fitness (W), in order to quantify the rate of decrease of fitness while increasing the purged inbreeding coefficient by one unit (i.e., from zero to one), the calculated value of δ has to be raised to the power of the mathematical constant (approximately equal to 2.718) then it has to be multiplied by the expected fitness of the non-inbred individuals estimated from the inbreeding-purging model. In our case, it is $e^{(\delta \times g)} \times W_0 = 2.718^{(-0.442081 \times 1.0)} \times 0.931144 = 0.5984$. Thus, the fitness of an animal having a purged inbreeding coefficient of one is 35.7% lower than that of the non-inbred animals. Regarding the other factors, the fitness of rabbit kits born in summer was inferior compared to the rest of the year while latter parities were favorable when compared to the first kindling in both periods (Table 1).

The estimated effective population size was 114 with a standard error of 1.26, with fluctuations observed throughout breeding from 1992 to 2017. The largest fluctuation in effective population size began with an increase in the number of breeding rabbits in 1993, followed by a sudden decrease in 1998, as described in Nagy et al. [33] and Curik et al. [24]. Thus, the effective population size was estimated to be over 250 in 1995 [33]. Additionally, the survival rate at birth also exhibited fluctuation between 1992 and 2017 where after the decreasing tendency, which lasted for few years, a clear increasing tendency could be observed [24]. The evolution of the conventional (line in red) and purged (line in blue) inbreeding coefficients together with the successive years are depicted in Figure 1 using the purging coefficient estimated for the period of 1992–1997 based on the heuristic approach of García-Dorado et al. [30]. The darker colored dots indicate greater data frequency. At

the beginning the trend of the two types of inbreeding, coefficients could not be separated but after around 10 years (which is approximately 10 generations), the trend of the purged inbreeding coefficient exhibited a tendency to increase slower. By the end of the examined period, the median values of the Wright and the purged inbreeding coefficients were 0.107 and 0.044, respectively. In Figure 1, there are some dots around the value of 0.250, which clearly indicate some unintentional matings of some relatives, but the frequency of such matings was low.

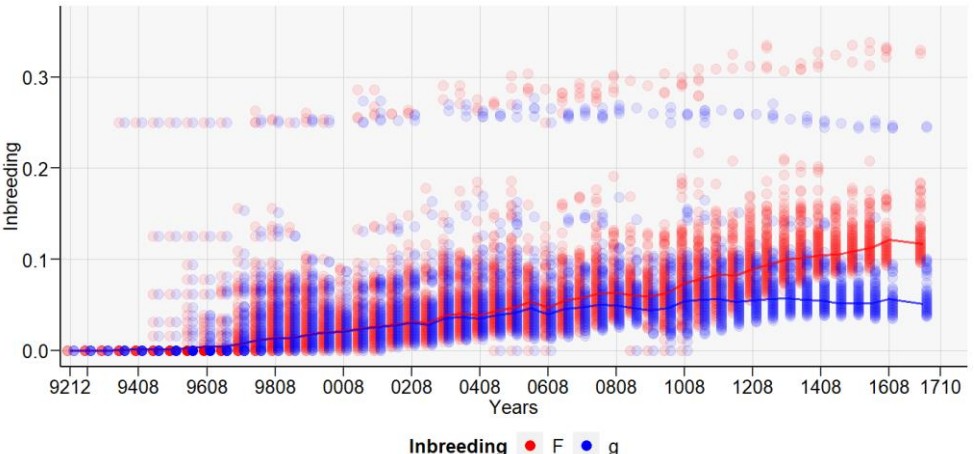

**Figure 1.** Trends of the Wright (F) and the purged (g) inbreeding coefficients throughout the examined period of 1992–2017.

The expected fitness estimated from the parameters provided in Table 1 is presented in Figure 2. Looking at the trends based on the Wright and on the purged inbreeding coefficients, it can be seen that in general the predicted fitness of the population was somewhat lower at the end of the examined period compared to the starting value. However, the magnitude of the decrease was not large regardless of the considered inbreeding coefficients' type. Nevertheless, similar to the inbreeding trend, the predicted fitness values started to separate after 10 generations. The fitness trend based on the Wright coefficient showed a small continuous decrease until the end of the examined period. On the contrary, the fitness trend based on the purged inbreeding stabilized between the 10th and 20th generations and did not show a further decrease afterward. Nevertheless, the observed difference between the two predicted fitness values at the end of the study was not extremely large (0.894 vs. 0.912).

The expressed opportunity of purging of the Pannon White rabbit population and the complementary analysis based on the purged inbreeding are presented in Figures 3 and 4. Looking at the presented trend of the expressed opportunity of purging, the inbreeding load started to decrease only after 10 generations and by the end of the analyzed period the inbreeding load was between 40% and 80% of its original value, meaning that the decrease of the load was at least 20% of the original magnitude (Figure 3). On the contrary, when the calculation was based on the purged and unpurged inbreeding coefficients, the decrease of the load was negligible when the purging coefficient was considered to be zero (Figure 4).

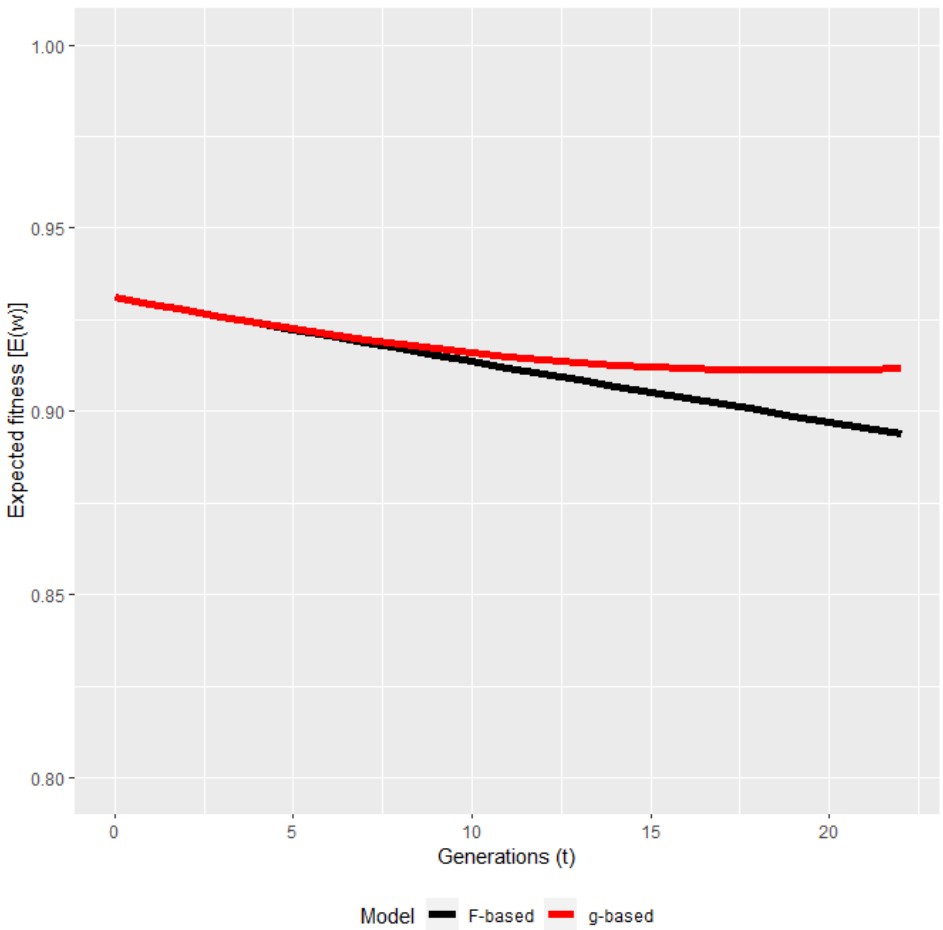

**Figure 2.** The predicted fitness based on the Wright (F) and on the purged (g) inbreeding coefficients.

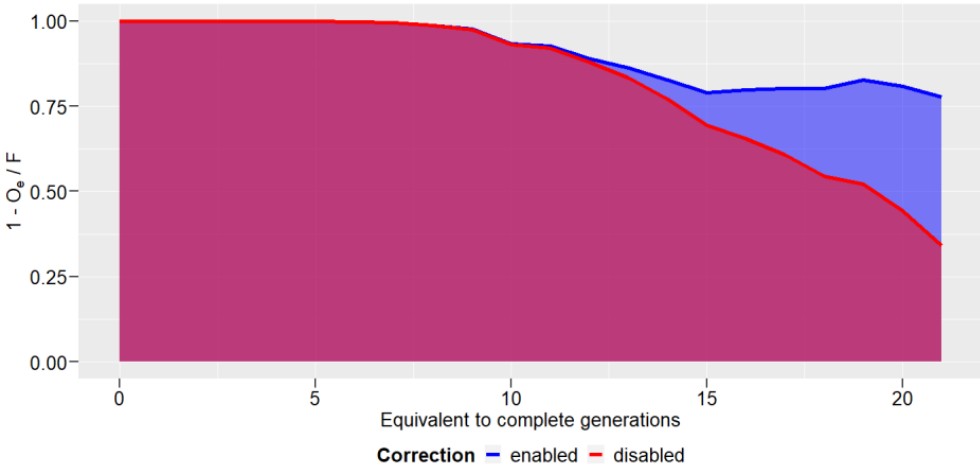

**Figure 3.** Expressed opportunity of purging.

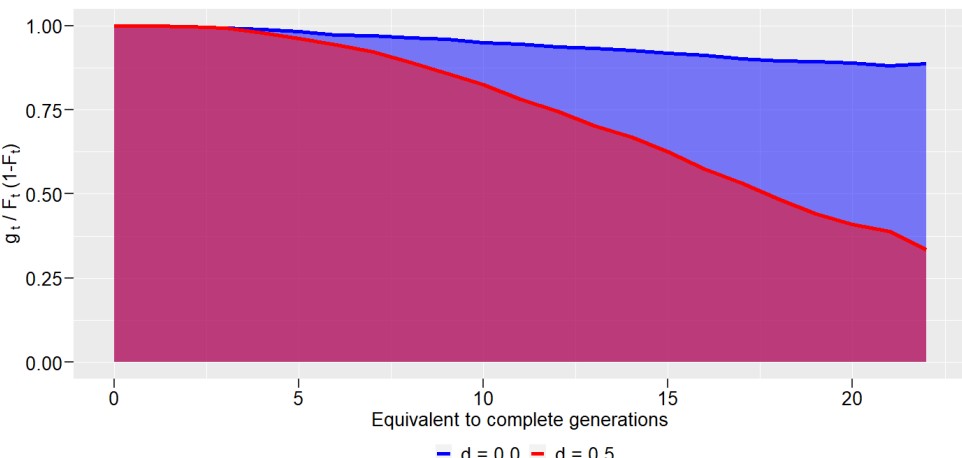

**Figure 4.** The proportional reduction of the inbreeding load based on García-Dorado [27].

## 4. Discussions

According to the ancestral inbreeding theory, one of the signals for the possible existence of purging is obtaining significantly positive effects of the ancestral inbreeding coefficient (or its interaction with inbreeding coefficient) for the examined traits [14,20]. Obtaining such results in domesticated species is rare and it was reported in few studies [24–26]. Our present study examining inbreeding depression observed that during the first period, the increasing litter inbreeding significantly decreased the survival rate at birth (Table 1). The significant negative litter inbreeding effects observed in the first period turned to significantly positive effect when the whole period was considered (Table 1). Our previous study [24] analyzing the same dataset in the first period (1992–1997) demonstrated that the litter inbreeding negatively affected the number of kits born alive while the Kalinowski ancestral inbreeding coefficient had a significantly positive effect on that trait. However, after 1997 neither the Wright nor the ancestral inbreeding affected the kits survival at birth. Very similar tendencies were reported in a captive gazelle population [34] where after the first period with inbreeding depression, increased juvenile survival (until the age of 14 days) coincided with increasing inbreeding coefficients in accordance with the signals of purging. Although not related to animal science, child survival of the Habsburg dynasty also behaved in a very similar manner where the inbreeding load experienced a strong reduction from the first to the second period in 10 generations [35]. Besides the ancestral inbreeding, signals of purging can also be detected based on the inbreeding-purging model [27]. From comparing the ancestral inbreeding theory and the inbreeding-purging model with computer simulations, it can be seen that the most favorable characteristic of the inbreeding-purging model is that it makes the unbiased prediction of the evolution of mean fitness in populations possible [28]. The possibility of unbiased fitness prediction of the Pannon White rabbit population was one of the main reasons for undertaking our present study. The possibility of purging using the inbreeding-purging model is based on the product of the purging coefficient (d) and the effective population size where their product must exceed one [36]. The large value of the purging coefficient presented in Table 1, based on the heuristic approach of García-Dorado et al. [30], suggests that the initial inbreeding load of the Pannon White rabbit population was at least partly caused by severe deleterious alleles that were purged from the population. It must be emphasized that the statistical analysis of whether the purging coefficient is equal to zero was not significant, which contradicts the results observed in other analyses. At the same time, we do not know to what extent the large change in effective population size observed in the first period might affect the sensitivity of our results and statistical conclusions. In captive G. cuvieri (d = 0.48) and N. dama (d = 0.23) ungulate populations, [36] reported that the large d estimates suggested that a substantial fraction of the initial inbreeding load was purged under the modest (11–14) effective population size. Other estimates for purging coeffi-

cients were reported for Drosophila fitness traits kept in laboratory conditions (d = 0.09 [22], d = 0.3 [23]). The trend of purged inbreeding level and predicted fitness in the population of Pannon White rabbits stabilized after about 15 generations and showed no further decline (Figures 1 and 2). This indicates a partial purging in which highly deleterious alleles were successfully eliminated while mildly deleterious alleles were fixed [18]. The stabilizing fitness as a consequence of purging was even more apparent for N. dama [36], where after the initial decrease a fitness rebound was detected during the last couple of generations. In accordance with the results presented above, the initial inbreeding load of the Pannon White rabbit population at the end of the examined period was between 40% and 80% of its original value (Figure 3): a result which also supports purging. Similar results were reported in a population of Jersey cattle [37], but to our knowledge this was the only case in which this type of analysis was performed in domesticated species. However, based on the expressed opportunity of purging, the reported magnitude of the inbreeding load reduction was smaller (12.6%) in the Jersey population than in our study. On the contrary, when the purging coefficient was considered to be zero, the estimated decrease of the genetic load was negligible. Nevertheless, as noted by Bundgaard et al. [38], the environment is crucial from the aspect of purging since inbreeding depression is positively correlated with the stressfulness of the environment. Furthermore, it has to be noted that our circular mating scheme may have contributed to slow purging for survival, as it implies some equalization of family contributions and, therefore, some relaxation in natural selection, as it was demonstrated by Pérez-Pereira et al. [39] based on a computer simulation study.

It is known that inbreeding depression is more pronounced when environmental conditions are more stressful [38]. Therefore, we cannot exclude the possibility that severe purging in domestic animals is statistically difficult to detect because environmental conditions on farms are so favorable that inbreeding depression is never fully expressed. This may be the case in our analysis, in which signs of purging, although either suspected (in this study) or previously observed [24], were not statistically confirmed, i.e., the purging coefficient was not significantly different than zero.

## 5. Conclusions

The results of the analyses performed in this study of the potential purging of negative inbreeding effects on survival at birth of Pannon White rabbits by different approaches ("inbreeding-purging" and "expressed opportunity to purge") did not fully agree because we were not statistically able to reject the hypothesis that the purging coefficient was different than zero. Thus, our study additionally shows that it is very difficult to detect purging statistically, especially in domestic animal populations where the environment is mostly favorable and inbreeding depression is not fully pronounced. On the other hand, we have made a number of arguments that, together with our previous results, suggest the presence of mild purging.

**Author Contributions:** Conceptualization, G.K., I.C. and I.N.; data curation, J.F., G.K. and L.V.; formal analysis, J.F. and G.K.; funding acquisition, I.N. and D.M.; investigation, I.N.; methodology, G.K., I.C. and I.N.; project administration, I.N.; resources, D.M.; software, I.N. and G.K.; supervision, I.N. and L.V.; validation, G.K., J.F. and I.N.; visualization, G.K., I.C. and I.N.; writing—original draft, I.N.; writing—review & editing, G.K. and I.N. All authors have read and agreed to the published version of the manuscript.

**Funding:** This research is funded by the OTKA 128 177 project (NKFIH).

**Institutional Review Board Statement:** Animal Care and Use Committee approval was not obtained for this study, because analyses were carried out on recorded data collected under standard farm management. These records were handled by the head of the experimental farm of the Kaposvár University (predecessor of the Hungarian University of Agriculture and Life Sciences) between 1992 and 2017.

**Data Availability Statement:** The data presented in this study are available on request from the corresponding author.

**Conflicts of Interest:** There were no conflict of interest among authors.

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
