# Peer review of "Analysis of Inbreeding Effects on Survival at Birth of Pannon White Rabbits Using the Inbreeding-Purging Model"

_diversity, doi:10.3390/d15010071_

Round 1
Reviewer 1 Report
This manuscript analyzes the joint effect of inbreeding and purging in a captive breed of rabbits. There are few examples of such analysis and this one is based in a valuable data set and can provide relevant information to understand the genetic architecture of inbreeding depression and the role of purging in the evolution and survival of small populations. However, there is a number of important questions and concerns that should be addressed or clarified before it can be published. I will mention first the more general concerns and then some specific comments
General comments:
A) To begin with the issue that first catches the reader attention, the references in the text and the literature section are a terrible mess. Regarding the sources cited, the manuscript only quotes evidence corresponding to the livestock literature, but inbreeding depression and purging are general processes for which evidence has accumulated for many decades and even centuries. Not that an extensive quoting is affordable, but there are reviews that can be cited to settle the context properly. From the formal point of view, the literature section contains a mixture of standard references and doi numbers. The dois are often repeated several times. At some point, each reference is followed by its doi number in the next line, and both are assigned different numbers. I am not making comments on the adequacy of the places were the references are included in the text because I think they are not reliable. I guess some reference editor has been used and the result has not been checked. Pleas, check carefully.
B) Addressing now the paper’s contribution, some information should be given to allow a more useful discussion. I don´t think and exhaustive demographic analysis is necessary or helpful, but some information should be given on the origin of funder individuals so that it can be inferred how likely their genomes can have undergo inbreeding and purging in the past. Also, we need some information on the evolution of the average population numbers and the survival rates through generations. I think it would be better including such figures in the ms but, if they are not included, Figure 1 in Curik et al. 2020 should be cited (where survival is average each month, right?). According to Fig 1 in the present manuscript, the increase of inbreeding is roughly linear, as expected for relatively large and constant effective size Ne. The average inbreeding observed at the end of the period seems to be about 0.1 which, assuming 23 generations and using the classical Wright´s expression for the evolution of F, could be explained by a constant effective size Ne=109. A result of this kind would be useful to get a rough idea about the magnitude of the recessive components of deleterious effects that are expected to be efficiently purged. Summary information on the population management would also be helpful (e.g., whether breeding contributions where random or were equalized or optimized to some extent and whether close inbreeding was avoided when individuals were mated).
C) General interpretation: The I-P model gives predictions as a function of the inbreeding load and the purging coefficient of the initial population (d and d, respectively), and these are the parameter that are estimated by using this model. Thus, the method does not estimate the average d and d values through the analyzed period, but the values in the initial reference population that is considered non-inbred. It is true that, if the estimates are obtained from a long period where purging occurred only during a small initial fraction, the estimates of d (and of d) tend to be downwardly biased (García-Dorado et al. 2016), so that estimates obtained in the first period, if significant, are more reliable. But 24 generations represent a relatively short period, taking into account the effective population size.
D) Taking into account the previous comment, it is unclear why there is such a large difference between the estimates obtained during the different periods. In fact, looking at figure 1, the d=0.5 estimate was obtained with data from an initial period during which d=0.5 leads to very similar predictions for F and g, which is not expected to allow d estimation. A first question is therefore how far these d estimates are significantly different from zero. This is tested by the approximate CHI2 test in the PURGd output (and, I assume, in purgR), but is not reported. I have extensively checked the different versions of PURGd, and I recently detected some errors in the statistical results of the output of PURGd v2.3.2. that affect AICc and CHI2 values. Usually the consequences are minor, but they can occasionally be relevant. I don’t know if these bugs have been fixed in purgR, but considering that the same guitlab that offers purgR still offers PURGd v.2.3.2, I think it should be convenient to check. This can done using The PURGd version in my repository: https://www.ucm.es/gfm/mecanismos-geneticos
Which includes a user manual warning about these errors and an excel patch to compute correct AICc and CHI2 values.
E) Note that, although founding individuals are surviving individuals, they should not be assigned a survival=1 value, as survival is defined for newborns, not for adults. Usually, founding individuals are the only individuals available in generation 0 and have been incorporated because they survived to the reproductive stage. Therefore, assigning them survival=1 implies assigning 1 to the average survival of generation 0 while, most likely, many individuals born in the same generation died before reproductive age. This is a trivial comment and I hope it does not annoy the authors, but it is not impossible to forget some detail, and this one can lead to overestimating d and d.
F) Note also that, regarding inbreeding depression, the model is W=W0 exp(b g) or W=W0 exp(-d g). This means that d = -b (the sign of the nonlinear regression coefficient on g should be changed to obtain the d value). In lines 133 and 134, it is mentioned that inbreeding has a negative effect on survival in the first period and a positive one in the second. This suggests that the authors just forgot to change the sign in the table. Please check this critical issue. In this report I assume the sigh of d (and of dM) needs to be changed in the Tables.
G) Regarding the analysis performed over the whole period, d and dM have opposing signs, which is wired, although all the d and (mainly) dM estimates are very small. The first think we need to know is whether each d (or dM) estimate is significantly different from zero. PURGd allows to do this test by comparing the LogLikelihoods of two analysis that include the same factors except in that in one model d is estimated and in the other one d is assigned a zero value. This comparison can be done using the CHI2 approach, as in the patch provided in PURGd v.2.3.2. The authors give results from an ANOVA analysis that I guess should be equivalent, but they do not explain anything about how it is performed, and I find really surprising getting a so large significance for such small d estimates.
H) Regarding the evolution of the inbreeding load, the Gulisija and Crow approach accounts for the purging occurred against quasilethal recessive alleles. I suggest obtaining complementary predictions using the I-P model for the estimate of the d value (see Eqs 9 and 11 in García-Dorado 2012 although, as in the Gulisija and Crow, only the evolution of the inbreeding load due to alleles segregating in the original population can be predicted on the basis of the estimated parameter, so that Eq 9 is appropriate).
Specific comments:
Line 84: This is not Morton et al. equation (those authors dealt with F, not with g). Rather, it is Eq 23 in García-Dorado et al. 2016. Similarly, the heuristic approach mentioned in lines 92-93 is suggested and implemented in García-Dorado et al. 2016, where PURGd is presented. As far as I know, López- Cortegano 2021 presents an R package that includes an R version of PURGd v2.3.2.
Line 89: Better “adjusted by the expected frequency of the purged deleterious alleles relative to their value before purging”
Line 94: Actual d values are unknown. Maybe better: “For each d value assumed, this approach computes the gi value of each individual”.
Lines 96-98: I do not understand this. Does this mean “to obtain an estimate of the initial (non-inbred) fitness”? PURGd v.2.3.2 provides an option (--W0) to estimate W0 jointly with the remaining parameter using the non-linear -regression approach, which has been tested and produces reliable conservative estimates of d and d. This option should be used.
Lines 110-111: An additional estimate of the initial fitness (W’0) is obtained with individuals having g=0 and gm=0, but this is not fully appropriate, as they can have inbred ancestors where purging could have occurred (Fa>0), so that their expected fitness can be larger than that of the original non-inbred non-purged individuals. The additional estimate of W0 should be computed using individuals with F=Fa=0 (both for individuals and, if maternal inbreeding is included, for their mothers). The default option of the program uses this W’0 estimate as initial fitness value, which reduces the downward bias of d and d estimates but is only reliable if the standard error of W’0 is very small. It is likely that the estimate of W0 used makes no relevant difference in this case, but I think it is better to check.
Tables:
-How can the standard error of W’0 be so small?
-I guess d (p) etc means “d (p-value)”, where the p-value is for the null hypothesis d=0 (one or two-tailed test?). Please explain this in the Table footnotes. How are these p-values obtained. Note that (as explained in the user manual of PURGd v2.3.2) the standard deviations that are shown below the estimates in the PURGd output when the program is run n times (--nruns=n option; I did not run the R version), are just a measure of the quality of the convergence in the numerical approach. A non-numerical estimation method would produce always the same estimate from the same data, but that does not mean that the corresponding estimate would have no standard error. The standard error should include the component ascribed to the sampling error, which produces differences between estimates obtained from different data. The PURGd approach does not give standard errors for the estimates. Approximate tests for their significance can be obtained by comparing the loglikelihoods of analysis where the parameter has been estimated or has been ascribed a zero value. By the way, it would be convenient to mention the number of runs used.
-As commented above, it is necessary to include in the Table the CHI2 values corresponding to the test for d=0. It can also be useful to provide in a sup. Mat. the results for the reference d=0 output corresponding to each analysis,
I look forward to see the revised version of this valuable piece of research where, hopefully, the results will allow a more enlightening discussion.
All the best
Author Response
Response1
This manuscript analyzes the joint effect of inbreeding and purging in a captive breed of rabbits. There are few examples of such analysis and this one is based in a valuable data set and can provide relevant information to understand the genetic architecture of inbreeding depression and the role of purging in the evolution and survival of small populations. However, there is a number of important questions and concerns that should be addressed or clarified before it can be published. I will mention first the more general concerns and then some specific comments
Dear Reviewer,
First of all, the authors would like to thank for your time spent on this very extensive review. We are pleased to agree with the reviewer the used dataset has certain characteristics which makes it very suitable for inbreeding-purging related analysis. We read thoroughly your comments and suggestions and provide a point by point response below.
General comments: A) To begin with the issue that first catches the reader attention, the references in the text and the literature section are a terrible mess. Regarding the sources cited, the manuscript only quotes evidence corresponding to the livestock literature, but inbreeding depression and purging are general processes for which evidence has accumulated for many decades and even centuries. Not that an extensive quoting is affordable, but there are reviews that can be cited to settle the context properly. From the formal point of view, the literature section contains a mixture of standard references and doi numbers. The dois are often repeated several times. At some point, each reference is followed by its doi number in the next line, and both are assigned different numbers. I am not making comments on the adequacy of the places were the references are included in the text because I think they are not reliable. I guess some reference editor has been used and the result has not been checked. Pleas, check carefully.
We have to admit that when placing the doi numbers it skipped our attention that at several places we put the doi into a new line making the reference numbering completely wrong. This error has been corrected and the doi numbers are also checked. Besides, we acknowledge that the references were restricted to livestock related studies. According to the reviewer’s suggestion the cited references were extended both in time frame and in covering a more broad scientific field.
- B) Addressing now the paper’s contribution, some information should be given to allow a more useful discussion. I don´t think and exhaustive demographic analysis is necessary or helpful, but some information should be given on the origin of funder individuals so that it can be inferred how likely their genomes can have undergo inbreeding and purging in the past. Also, we need some information on the evolution of the average population numbers and the survival rates through generations. I think it would be better including such figures in the ms but, if they are not included, Figure 1 in Curik et al. 2020 should be cited (where survival is average each month, right?). According to Fig 1 in the present manuscript, the increase of inbreeding is roughly linear, as expected for relatively large and constant effective size Ne. The average inbreeding observed at the end of the period seems to be about 0.1 which, assuming 23 generations and using the classical Wright´s expression for the evolution of F, could be explained by a constant effective size Ne=109. A result of this kind would be useful to get a rough idea about the magnitude of the recessive components of deleterious effects that are expected to be efficiently purged. Summary information on the population management would also be helpful (e.g., whether breeding contributions where random or were equalized or optimized to some extent and whether close inbreeding was avoided when individuals were mated).
We have to admit that detailed demographic analysis was not our intention since in one of our earlier study (Nagy et al., 2010) a very detailed demographic analysis was performed for the same rabbit population. This work is now also cited. According to the suggestion figure 1 of Curik et al. is referred. As the effective population size is used for several figures, it is explicitly mentioned that the effective population size estimated for the whole period was 114. Besides, we confirm that a so called circular mating scheme is applied in order to control the inbreeding rate. The basic element of this method has been added to the revised manuscript.
- C) General interpretation: The I-P model gives predictions as a function of the inbreeding load and the purging coefficient of the initial population (d and d, respectively), and these are the parameter that are estimated by using this model. Thus, the method does not estimate the average d and d values through the analyzed period, but the values in the initial reference population that is considered non-inbred. It is true that, if the estimates are obtained from a long period where purging 2 occurred only during a small initial fraction, the estimates of d (and of d) tend to be downwardly biased (García-Dorado et al. 2016), so that estimates obtained in the first period, if significant, are more reliable. But 24 generations represent a relatively short period, taking into account the effective population size.
This comment is acknowledged.
- D) Taking into account the previous comment, it is unclear why there is such a large difference between the estimates obtained during the different periods. In fact, looking at figure 1, the d=0.5 estimate was obtained with data from an initial period during which d=0.5 leads to very similar predictions for F and g, which is not expected to allow d estimation. A first question is therefore how far these d estimates are significantly different from zero. This is tested by the approximate CHI2 test in the PURGd output (and, I assume, in purgR), but is not reported. I have extensively checked the different versions of PURGd, and I recently detected some errors in the statistical results of the output of PURGd v2.3.2. that affect AICc and CHI2 values. Usually the consequences are minor, but they can occasionally be relevant. I don’t know if these bugs have been fixed in purgR, but considering that the same guitlab that offers purgR still offers PURGd v.2.3.2, I think it should be convenient to check. This can done using The PURGd version in my repository: https://www.ucm.es/gfm/mecanismos-geneticos Which includes a user manual warning about these errors and an excel patch to compute correct AICc and CHI2 values.
Authors would like to thank the reviewer for this hint of advice. In fact, the purgeR manual did not contain this CHI2 test, only the heuristic approach which was applied by the authors. We emphasize again that for the first period (1992-1997) the smallest AIC value was found for d=0.5 although it has to be mentioned that the AIC between d=0 and d=0.5 was minimal. Nevertheless, based in the given information the mentioned CHI2 test was performed and the related p values are added to table 1. According the results p values were larger than 0.05 thus d=0 could not be rejected. This suggests that the heuristic approach and the CHI2 test gave discordant results.
- E) Note that, although founding individuals are surviving individuals, they should not be assigned a survival=1 value, as survival is defined for newborns, not for adults. Usually, founding individuals are the only individuals available in generation 0 and have been incorporated because they survived to the reproductive stage. Therefore, assigning them survival=1 implies assigning 1 to the average survival of generation 0 while, most likely, many individuals born in the same generation died before reproductive age. This is a trivial comment and I hope it does not annoy the authors, but it is not impossible to forget some detail, and this one can lead to overestimating d and d.
Authors confirm that founding animals were not assigned with survival=1.
- F) Note also that, regarding inbreeding depression, the model is W=W0 exp(b g) or W=W0 exp(-d g). This means that d = -b (the sign of the nonlinear regression coefficient on g should be changed to obtain the d value). In lines 133 and 134, it is mentioned that inbreeding has a negative effect on survival in the first period and a positive one in the second. This suggests that the authors just forgot to change the sign in the table. Please check this critical issue. In this report I assume the sigh of d (and of dM) needs to be changed in the Tables.
We confirm that as the reviewer noted we simply forgot the change the sign in the table. This mistake has been corrected.
- G) Regarding the analysis performed over the whole period, d and dM have opposing signs, which is wired, although all the d and (mainly) dM estimates are very small. The first think we need to know is whether each d (or dM) estimate is significantly different from zero. PURGd allows to do this test by comparing the LogLikelihoods of two analysis that include the same factors except in that in one model d is estimated and in the other one d is assigned a zero value. This comparison can be done using the CHI2 approach, as in the patch provided in PURGd v.2.3.2. The authors give results from an ANOVA analysis that I guess should be equivalent, but they do not explain anything about 3 how it is performed, and I find really surprising getting a so large significance for such small d estimates.
As mentioned above the CHI2 tests were performed according to the given website and according to their results d=0 could not be rejected. The mentioned ANOVA analysis was used to compare two model with and without δ, however, after merging tables 1 and 2 this part has been removed from the manuscript.
- H) Regarding the evolution of the inbreeding load, the Gulisija and Crow approach accounts for the purging occurred against quasilethal recessive alleles. I suggest obtaining complementary predictions using the I-P model for the estimate of the d value (see Eqs 9 and 11 in García-Dorado 2012 although, as in the Gulisija and Crow, only the evolution of the inbreeding load due to alleles segregating in the original population can be predicted on the basis of the estimated parameter, so that Eq 9 is appropriate).
Thank you very much this suggestion which was followed and a new figure was added (Figure 4) based on equation 9 of the study published by Dorado (2012).
Specific comments: Line 84: This is not Morton et al. equation (those authors dealt with F, not with g). Rather, it is Eq 23 in García-Dorado et al. 2016. Similarly, the heuristic approach mentioned in lines 92-93 is suggested and implemented in García-Dorado et al. 2016, where PURGd is presented. As far as I know, LópezCortegano 2021 presents an R package that includes an R version of PURGd v2.3.2.
The cited references were changed according to the suggestions.
Line 89: Better “adjusted by the expected frequency of the purged deleterious alleles relative to their value before purging”
This suggestion has been accepted.
Line 94: Actual d values are unknown. Maybe better: “For each d value assumed, this approach computes the gi value of each individual”.
This suggestion has been accepted.
Lines 96-98: I do not understand this. Does this mean “to obtain an estimate of the initial (non-inbred) fitness”? PURGd v.2.3.2 provides an option (--W0) to estimate W0 jointly with the remaining parameter using the non-linear -regression approach, which has been tested and produces reliable conservative estimates of d and d. This option should be used.
Since the purgeR was used for the calculation we followed the workflow suggested by the vignettes which are part of the purgeR help system. Using that workflow all the parameters including the W0 of the nonlinear formula
were estimated by the stats::nls function of the R. This function serves to determine the estimates of the parameters of a nonlinear model. This nls function requires starting values of the estimates. The starting values of the parameters were determined by a previously executed probit regression.
So the “initial” word in this context was used to refer to the starting values of a nonlinear model.
The vignettes:
Eugenio López-Cortegano, 2022-07-11 Inbreeding and Purging Estimates
Eugenio López-Cortegano, 2022-07-11 purgeR tutorial
Lines 110-111: An additional estimate of the initial fitness (W’0) is obtained with individuals having g=0 and gm=0, but this is not fully appropriate, as they can have inbred ancestors where purging could have occurred (Fa>0), so that their expected fitness can be larger than that of the original noninbred non-purged individuals. The additional estimate of W0 should be computed using individuals with F=Fa=0 (both for individuals and, if maternal inbreeding is included, for their mothers). The default option of the program uses this W’0 estimate as initial fitness value, which reduces the downward bias of d and d estimates but is only reliable if the standard error of W’0 is very small. It is likely that the estimate of W0 used makes no relevant difference in this case, but I think it is better to check.
Thank you for pointing out this mistake. It was simply a typing error from our part. The used script was correct, still we apologize for not being careful enough.
dplyr:: filter(Fi == 0 & Fa == 0) %>%
u2$Fi <- with(ped, Fi[match(u2$LITTERID,id)])
u2$Fa <- with(ped, Fa[match(u2$LITTERID,id)])
Tables: -How can the standard error of W’0 be so small?
Our kindling dataset contains 22718 kindling records with the number of kits born alive and dead. These kindling record were multiplied according to the number of the kits total born. The record numbers of the resulted “individual” kindling dataset contains 203065 records. F=Fa=0 individuals were used to calculate the descriptive statistics of W’0. In the first time period 36632 individual and 42236 in the whole period.
4 -I guess d (p) etc means “d (p-value)”, where the p-value is for the null hypothesis d=0 (one or twotailed test?). Please explain this in the Table footnotes. How are these p-values obtained. Note that (as explained in the user manual of PURGd v2.3.2) the standard deviations that are shown below the estimates in the PURGd output when the program is run n times (--nruns=n option; I did not run the R version), are just a measure of the quality of the convergence in the numerical approach. A nonnumerical estimation method would produce always the same estimate from the same data, but that does not mean that the corresponding estimate would have no standard error. The standard error should include the component ascribed to the sampling error, which produces differences between estimates obtained from different data. The PURGd approach does not give standard errors for the estimates. Approximate tests for their significance can be obtained by comparing the loglikelihoods of analysis where the parameter has been estimated or has been ascribed a zero value. By the way, it would be convenient to mention the number of runs used.
In the submitted version of our paper d (p) was not included. We couldn’t find the way how to calculate d(p) using the purgeR. However, after careful studying your opinion we contacted Eugenio López-Cortegano and followed his advice to obtain the Chi2 value and the p value inside the R environment. The formula we used is as follows:
Chi2 = AIC(model with d=0)-AIC(model with the best d)+1, df=1, lower.tail=FALSE.
The revised paper now contains the p-value of the Chi2 statistics in Table 1 and 2.
In Table 1 and 2 the estimates of the model parameters (W0, δ, δM, S2, P2, P3, P4) can be found. The R function stats::nls was used to estimate these values. In the output of this function not only the estimates can be found but also the standard error and the p-value of the t-test. (H0: parameter=0).
The purgeR has no “runs” option, we were not able to use a method like that.
-As commented above, it is necessary to include in the Table the CHI2 values corresponding to the test for d=0. It can also be useful to provide in a sup. Mat. the results for the reference d=0 output corresponding to each analysis,
Authors fully agree that performing the significance test for d=0 is having great importance. As it mentioned above these tests were performed and it is included in the revised manuscript.
I look forward to see the revised version of this valuable piece of research where, hopefully, the results will allow a more enlightening discussion.
Authors would like to thank again the extremely thorough review which definitely will improve the scientific level of the manuscript to a great extent. We look forward to receive the reviewer response to the revised manuscript.
With best regards,
Istvan Nagy
Reviewer 2 Report
Very interesting article concerning inbreeding purging model in a close and endangered rabbit population. Congratulations for your work.
Some minor comments and suggestions for improvement of the manuscript:
L55 - consider changing "..re-analyse to dataset..." to "... re-analyse the dataset..."
L69 - please try to explain the implications and why the objectives of the breed have changed several times
L95 reference to purgeR and following references to software packages should have a citation/reference associated and a version indicated.
L97 - reference for R software L101 - reference to AIC package
L101 - clarify and indicate what means AIC first time mentioned
L104 - why only 2 seasons considered and not 4?
L122 - explanation for the reason of 2 periods of time so different (5 vs 15 years long?)
L127 - space missing "Tables 1_and 2"
Tables 1 and 2 - p(Anova) format, improve editing. Missing legend for "NA"
legend for values ( )
Table 1 cannot be in 2 pages
Figures 1 and 3 - too small text in legend of axis XX and YY, cannot read. Improve editing and format.
L223 - ref 21 not according with Habsburg citation. About German Holstein cattle. Revise all references to be according with citations in the entire manuscript.
L240 - cca maybe latin ca? "around2?
References with several incomplete data. For example in ref numbers 6, 7, 9, 11, etc only with DOI and no authors and complete citation. Revise all reference chapter.
Author Response
Response2
Very interesting article concerning inbreeding purging model in a close and endangered rabbit population. Congratulations for your work.
Authors would like to thank the reviewer for the favourable opinion. We also hope that our work may contribute to the genaral knowledge of inbreeding and purging.
Some minor comments and suggestions for improvement of the manuscript:
L55 - consider changing "..re-analyse to dataset..." to "... re-analyse the dataset..."
The suggestion is accepted and the text is corrected accordingly.
L69 - please try to explain the implications and why the objectives of the breed have changed several times
From the early 1990s the breed was selected for average daily gain and for the cross-sectional area of the m. Longissimus Dorsi measured at two anatomical points the so caloled L-value. Then the L-value has been replaced by thigh muscle volume because it covered a large proportion of the valuable parts of the carcass. Due to the intensive selection the rearing ability of the does decrased which initiated yet another change and average daily gain has been replaced by litter weight at day 21.
L95 reference to purgeR and following references to software packages should have a citation/reference associated and a version indicated.
Reference to purgeR is given but even the author López-Cortegano (2022) did not specify version number.
L97 - reference for R software L101 - reference to AIC package
R version 4.2.1
R Core Team (2022). R: A language and environment for statistical
computing. R Foundation for Statistical Computing, Vienna,
Austria. URL https://www.R-project.org/.
Mazerolle MJ (2020). AICcmodavg: Model selection and multimodel inference based on (Q)AIC(c). R package version 2.3-1, https://cran.r-project.org/package=AICcmodavg.
Authors are not fully convinced if these packages have to be explicitly cited so at the moment these citations are only listed here. In case the reviewer maintains that these (and other R) packages should explicilety be cited then the manuscript will be further extended.
L101 - clarify and indicate what means AIC first time mentioned
Authors admit that we simply forgot he define AIC at its first use. This mistake has been corrected.
L104 - why only 2 seasons considered and not 4?
Before the study of Curik et al. 2020 one of the co-authors Prof. Dr. Zsolt Szendrő (who is the developped the Pannon White rabbit breed advocated that only two seasons should be used the hot vs. non-hot). Since this study is a re-analysis of the work of Curik et al all effects ared defined in the same way as in that study.
L122 - explanation for the reason of 2 periods of time so different (5 vs 15 years long?)
The same explanation could be given as above. Looking figure 1. of Curik et al. it can be seen that in the survival rate of rabbit kits there was a decresing and increasing trend which was analyzed by separately analyzing two periods of 1992-1997 and 1997 and 2017. In this study we repeated the analysis only used different methodology.
L127 - space missing "Tables 1_and 2"
Due to some text editing we merged table 1 and table 2.
Tables 1 and 2 - p(Anova) format, improve editing. Missing legend for "NA"
legend for values ( )
The ANOVA analysis was used to compare two model with and without δ. This latter p value was however deleted from the manuscript after some editing while merging tables 1 and 2 into one table.
NA indicates where estimates were not applicable.
Table 1 cannot be in 2 pages
We accept the suggestion. The two tables were merged. The modified Table 1 is placed at one page.
Figures 1 and 3 - too small text in legend of axis XX and YY, cannot read. Improve editing and format.
Authors acknowledge that the legends of all figures were too small. In the revised manuscript all figures were updated. We hope that this change will help readability of the manuscript.
L223 - ref 21 not according with Habsburg citation. About German Holstein cattle. Revise all references to be according with citations in the entire manuscript.
As we also responded to our first reviewer we have to admit that when placing doi numbers it skipped our attention that at several places we put the doi into a new line making the reference numbering completely wrong. This error has been corrected and the doi numbers are also checked.
L240 - cca maybe latin ca? "around
Thank you for pointing out this mistake. The related text has been changed to „around”
References with several incomplete data. For example in ref numbers 6, 7, 9, 11, etc only with DOI and no authors and complete citation. Revise all reference chapter.
We apologize again for the refences section which was impossible to follow. The whole section has been modified.
Authors would like to thank the reviewer yet again for the helpful and favourable evaluation. We look forward to receive your opinion related to the revision of our manuscript.
With best regards,
Istvan Nagy
Reviewer 3 Report
Review Klöver et al., - Analysis of inbreeding effects on survival at birth of Pannon 2 White rabbits using inbreeding-purging model
Dear authors,
thank you for submitting an article on the important topic of inbreeding depression. You are running interesting analyses on a substantial dataset. It would be a valuable methodological contribution to the field of inbreeding related topics that will be in main focus in animal sciences in the following years, as the diversity crisis reaches the domestic breeds more and more.
I have to criticize that the submitted paper appears to be in a relatively preliminary state. The main contribution this paper promises is a new (actually old) approach to estimate inbreeding depression. This method is not trivial and not established in the field, to my knowledge and the authors´claims. Other findings regarding inbreeding depression are in good agreement with a prior study by Curik and therefore lack novelty. Given this, the introduction should include more aspects of methods and the theoretical background of inbreeding depression estimation, rather than the general address of inbreeding, inbreeding depression and purging. Consequently, the discussion would be expected to evaluate the benefits and drawbacks of the methodology, rather than discussing the estimated results, which are reported with a far simpler appearing and more intuitive method by Curik et al.. As both studies provide only estimates for ID and purging, it seems to be difficult to say which are the better ones. Especially the hypothetical assumption of large deleterious effects that are purged in the early phase of the population history seems to be weakly supported. It could be a good idea to benchmark the method on a simulated dataset first.
Some inaccuracies and overly complicated formulations need to be reviewed to provide better access to the readers. They are listed beneath:
General:
References need formatting, as the urls are listed as separate entries which makes also reviewing quite difficult. The style of writing could be improved in many places, as there is some potential to gain more readers interest by better readability. Some examples are listed in the minor comments.
Introduction
The introduction provides an overview of selected publications regarding inbreeding and purging. Most mentioned points follow population genetics theory and are widely accepted. It turns out at the end of the chapter that the scope of the paper is to introduce a novel statistic to quantify number and effect size of purged loci and therefor contribute to understand the genetic architecture of survival at birth. All this should be done in comparison with a former study by Curik et al.
With respect to such a specific scope, the introduction stays too general. Concepts of estimating load, purging and inbreeding depression must at least be briefly explained to show the conceptual difference of the novel approach. It would help to give an overview of former studies and load estimates on that trait.
Materials and Methods
2.1 needs better explanation of the dataset. It is unclear to me, even after revisiting Curik et al, what an artificial litter/ dummy progeny is and how you use it to extend the pedigree
2.2 the formula is not sufficiently broken down and not explicitly stated in Morton et al. Please improve the M&M section to a level which makes understanding the study possible without revisiting secondary literature that uses alternative labeling and nomenclature. e.g. initial values are determined by probit regression, but the formulation is missing. W is later labeled Wi, calculating Opportunity of purging , you introduce j an I while they are left undefined and so forth. Together with the lack of description of the dataset, I feel unable to understand thoroughly what you did. What is the actual phenotype you feed into the model and where is it actually used?
Also, Delta*G and DeltaM*gM, are dependent on each other, respectively. I understand that you are using the AIC as some kind of likelihood-like target parameter, but from superficial evaluation this seems to be some circuit closure. Could you provide a more detailed explanation of the whole framework and how this model does not tend to overfitting?
Results
Why are the analysed periods unequal in length or chosen to be like this exactly?
The plots are not sufficient in quality and labeling.
Minor:
21 Consider rephrasing: “Based on the expected in fitness over generations and on the expressed opportunity of purging the beneficial effects of purging could be expected after 10 generations where the proportion of the purged genetic load could be between 20 and 60%.”
30 would discard or specify “diversity” used with the only meaning of animal number or mating choices. Sufficient to say closed population.
50: Could not find this very term in the cited paper. Your paper would benefit from explaining inbreeding-purging vs ancestral inbreeding. “so-called inbreeding-purging method was recently proposed by García-Dorado [14]”
55 typo: re-analyze the (to) dataset of Curik et al.
68 (exemplarily): consider proofreading: missing punctuation hampers readability. You also tend to agglomerate words in sections, in this case “Since”.
71 “respectively” grammatically not necessary in this case. Would be indicated if you report additionally for example gain in a subordinate clause.
74 Numbers not in agreement with Curik et al.
75: “All records obtained originated with 1421 bucks and 5339 does, respectively” does that mean the numbers of founding bucks and does were 1421 and 5339, respectively? Please rephrase the dataset description. It is unclear at this point, if a kindling record is the individual value of every kid, or a litter and how artificial litter identity was extended. Information such as average litter size, number of founders, generations covered, average pedigree completeness etc. would be of highest interest when analyzing pedigree data.
157 negative estimates of S2 mean a decreased fitness in summer? I understood from line 104 (With regard to the season of kindling (summer or otherwise) and parity of the rabbit 104 doe (merged to four groups) a dummy variable (Season1=0; Season2=1) ) that S1 is summer and S2 is otherwise. Notation seems to be inconsistent for season and parity.
166 hard to see trends from the low quality figure with lines in same color as dots
230 “The possibility of purging using the inbreeding-purging model is based on the product of the purging coefficient (d) and the effective population size where their product must exceed 1.” Is this common knowledge? Otherwise explain or reference it.
I completely miss the comparison to Curik et al study on same dataset in discussion while mentioned in conclusion.
Author Response
Response3
Comments and Suggestions for Authors
Review Kövér et al., - Analysis of inbreeding effects on survival at birth of Pannon 2 White rabbits using inbreeding-purging model
Dear authors,
thank you for submitting an article on the important topic of inbreeding depression. You are running interesting analyses on a substantial dataset. It would be a valuable methodological contribution to the field of inbreeding related topics that will be in main focus in animal sciences in the following years, as the diversity crisis reaches the domestic breeds more and more.
I have to criticize that the submitted paper appears to be in a relatively preliminary state. The main contribution this paper promises is a new (actually old) approach to estimate inbreeding depression. This method is not trivial and not established in the field, to my knowledge and the authors´claims. Other findings regarding inbreeding depression are in good agreement with a prior study by Curik and therefore lack novelty.
First of all, the authors would like to thank the reviewer for the very thorough evaluation of our work. We also hope that the topic is interesting and it contributes to the present state of knowledge of inbreeding theory. Besides, we admit that inbreeding related studies have been conducted for centuries. On the contrary inbreeding-purging model (Dorado, 2012) is a relatively new method which has not yet been applied in animal science thus according to our opinion it is in fact a novelty. Curik et al. 2020 and the present study used completely different methods. Before conducting this study, it was uncertain if the results of the present study will be concordant or discordant with that of the study of Curik et al. 2020.
Given this, the introduction should include more aspects of methods and the theoretical background of inbreeding depression estimation, rather than the general address of inbreeding, inbreeding depression and purging. Consequently, the discussion would be expected to evaluate the benefits and drawbacks of the methodology, rather than discussing the estimated results, which are reported with a far simpler appearing and more intuitive method by Curik et al..
We believe that the introduction already contains the most important citations covering this request (e.g. bersabé et al., 2013; López-Coregano et al., 2016; López-Coregano et al., 2018).
As both studies provide only estimates for ID and purging, it seems to be difficult to say which are the better ones. Especially the hypothetical assumption of large deleterious effects that are purged in the early phase of the population history seems to be weakly supported.
It was not our objective to decide which method is better than the other. The favourable features of the inbreeding-purging methods were mentioned by López-Coregano et al., 2018. According to the suggestion of the reviewer the the hypothetical assumption of large deleterious effects that are purged in the early phase of the population history was removed from the manuscript.
It could be a good idea to benchmark the method on a simulated dataset first.
Authors highly appreciate this suggestion. Nevertheless, we maintain our opinion the order of the real data based and simulation based study should be reversed compared to this suggestion. Based on the results of this study we will develop many scenarios and we will perform the comparative analysis of the ancestral inbreeding based and the inbreeding-purging model based methods. At present based only on one dataset we believe that performing such a comparative analysis would not be scientifically sound.
Some inaccuracies and overly complicated formulations need to be reviewed to provide better access to the readers. They are listed beneath:
General:
References need formatting, as the urls are listed as separate entries which makes also reviewing quite difficult. The style of writing could be improved in many places, as there is some potential to gain more readers interest by better readability. Some examples are listed in the minor comments.
Authors apologize for their careless behavior. Due to our mistake the DOI numbers of several citations appeared at a new line making the reference numbering a complete mess. This error has been corrected and the DOI numbers has also been checked.
Introduction
The introduction provides an overview of selected publications regarding inbreeding and purging. Most mentioned points follow population genetics theory and are widely accepted. It turns out at the end of the chapter that the scope of the paper is to introduce a novel statistic to quantify number and effect size of purged loci and therefor contribute to understand the genetic architecture of survival at birth. All this should be done in comparison with a former study by Curik et al.
As mentioned above this suggestion will be covered in our next work where based on computer simulations the raised issue can properly dealt with.
With respect to such a specific scope, the introduction stays too general. Concepts of estimating load, purging and inbreeding depression must at least be briefly explained to show the conceptual difference of the novel approach. It would help to give an overview of former studies and load estimates on that trait.
Authors acknowledge the opinion of the reviewer. However, authors believe that our manuscript also fits this suggestion since the conceptual differences between ancestral inbreeding and inbreeding-purging is demonstrated in the manuscript. Nevertheless, as mentioned above, authors extended the introduction section in order to make it more broad both in time frame and in scientific area.
Materials and Methods
2.1 needs better explanation of the dataset. It is unclear to me, even after revisiting Curik et al, what an artificial litter/ dummy progeny is and how you use it to extend the pedigree
In our farm information system only the breeding animals have identity number. Thus the original pedigree contains only breeding animals. Since breeding animals are not selected from every litter, in order to guarantee that the litter inbreeding could be calculated for every litter, litter identities had to be created for every kingling record. The pedigree was extended with these identities (called dummy progeny).
2.2 the formula is not sufficiently broken down and not explicitly stated in Morton et al. Please improve the M&M section to a level which makes understanding the study possible without revisiting secondary literature that uses alternative labeling and nomenclature. e.g. initial values are determined by probit regression, but the formulation is missing.
Assuming W is survival, a binary variable, the probit model is:
Since the probit model is described in quite a few textbooks, and in this case, it is only a preliminary tool before the application of the main method i.e., the nonlinear model fitting we believe that it is not necessary to include the probit formula in the paper.
The values of W0, d, dm, S2, P2, P3, P4 estimated by the probit regression will create the starting values of the parameters of the nonlinear model.
W is later labeled Wi, calculating Opportunity of purging , you introduce j an I while they are left undefined and so forth. Together with the lack of description of the dataset, I feel unable to understand thoroughly what you did. What is the actual phenotype you feed into the model and where is it actually used?
Wi has been modified to W. Besides, according to one of the other reviewer the formula belongs to García-Dorado, A.; Wang, J.; López-Cortegano. Predictive model and software for inbreeding-purging analysis of pedigreed populations. G3-Genes Genom. Genet. 2016, 6, 3593-3601. https://doi.org/10.1534/g3.116.032425
The analyzed fitness trait was survival of the rabbit kits at birth (W) which is a binomial variable. Our kindling dataset contains 22718 kindling records with the number of kits born alive and dead. These kindling record were multiplied according to the number of the kits total born. The record numbers of the resulted “individual” kindling dataset contains 203065 records.
Also, Delta*G and DeltaM*gM, are dependent on each other, respectively. I understand that you are using the AIC as some kind of likelihood-like target parameter, but from superficial evaluation this seems to be some circuit closure. Could you provide a more detailed explanation of the whole framework and how this model does not tend to overfitting?
This is valid question. We performed an additional analysis and found that the litter and dam purged inbreeding coefficients were higly correlated (0.8). When collinearity was checked calculating generalized variance inflation factor (GVIF) based on our results the GVIF values were low for all model elements ranging between 1.00 and 1.50 in the first and 1.00 and 3.03 in the whole period. Thus it could be concluded that there is no problem with our model from the aspect of collinearity. This part however does not have to be mentioned in the manuscript as it is clear from table 1 that those models without dam purged inbreeding coefficients gave better fit (lower AIC) compared to models when dam purged inbreeding was included both in the first and in the whole period.
Results
Why are the analysed periods unequal in length or chosen to be like this exactly?
As it was stated before, this study was the re-analysis of the study of Curik et al. 2020 thus the periods used in that study were not altered.
The plots are not sufficient in quality and labeling.
Authors accept this criticism. All the figures are updated in order to improve readability.
Minor:
21 Consider rephrasing: “Based on the expected in fitness over generations and on the expressed opportunity of purging the beneficial effects of purging could be expected after 10 generations where the proportion of the purged genetic load could be between 20 and 60%.”
We accept that the sentence was too long. It was split to two parts in order to improve readability.
30 would discard or specify “diversity” used with the only meaning of animal number or mating choices. Sufficient to say closed population.
The text is changed according to the suggestion.
50: Could not find this very term in the cited paper. Your paper would benefit from explaining inbreeding-purging vs ancestral inbreeding. “so-called inbreeding-purging method was recently proposed by García-Dorado [14]”
The reviewer is kindly asked to specify which term was not found in the cited paper?
Authors believe that based on our manuscript the differences between the ancestral inbreeding and the inbreeding-purging method was clarified.
55 typo: re-analyze the (to) dataset of Curik et al.
The text has been modified.
68 (exemplarily): consider proofreading: missing punctuation hampers readability. You also tend to agglomerate words in sections, in this case “Since”.
Authors tried their best to write the manuscript in the best possible was. Besides, the text was checked by a native English speaker. Authors therefore believe the quality of the manuscript is sufficient from language aspects. Nevertheless, we hope that the modified manuscript has been further improved also from language aspects.
71 “respectively” grammatically not necessary in this case. Would be indicated if you report additionally for example gain in a subordinate clause.
Respectively has been deleted.
74 Numbers not in agreement with Curik et al.
Authors checked the related numbers and found that the number of kindling records and the size of the pedigree were 22.781 and 29802 in both studies.
Could you please clarify which parameter did you refer to?
75: “All records obtained originated with 1421 bucks and 5339 does, respectively” does that mean the numbers of founding bucks and does were 1421 and 5339, respectively?
No, it means that the number of different bucks are does on the whole pedigree file was 1421 and 5339. respectively.
Please rephrase the dataset description. It is unclear at this point, if a kindling record is the individual value of every kid, or a litter and how artificial litter identity was extended.
The text is extended clarifying the raised issue.
Information such as average litter size, number of founders, generations covered, average pedigree completeness etc. would be of highest interest when analyzing pedigree data.
In one of our previous work a detailed demographic analysis was performed (Nagy et al., 2010). It was not our intention to repeat these evaluations.
157 negative estimates of S2 mean a decreased fitness in summer? I understood from line 104 (With regard to the season of kindling (summer or otherwise) and parity of the rabbit 104 doe (merged to four groups) a dummy variable (Season1=0; Season2=1) ) that S1 is summer and S2 is otherwise. Notation seems to be inconsistent for season and parity.
This part has been modified.
166 hard to see trends from the low quality figure with lines in same color as dots
All figures have been updated
230 “The possibility of purging using the inbreeding-purging model is based on the product of the purging coefficient (d) and the effective population size where their product must exceed 1.” Is this common knowledge?
The manuscript has been extended with the requested citation.
I completely miss the comparison to Curik et al study on same dataset in discussion while mentioned in conclusion.
Curik et al. 2020 mentioned several times in the manuscript. It is a published open access article so there is no need to repeat its findinds more in detail. As mentioned before a detailed comparative analysis can only be performed having several datasets. In our planned computer simulation work this issue will be covered.
Finally, authors would like to thank the reviewer yet again the effort improving our manuscript.
With best regards,
Istvan Nagy
Round 2
Reviewer 1 Report
The authors did a good job reviewing the manuscript that is now clearer and more convincing, and I think makes a remarkable contribution to the field. I think the manuscript can be published after a few minor issues are addressed (see below).
Line 21: Replace with “reached the maximum possible value of 0.5 when estimated between 1992-1997”, as the method always estimates de initial parameter, although bias increase when too long periods are analysed. In the same line, in “expected in fitness”, remove “in”.
Line 76: remove “that”.
Line 143: “coincided with the lowest AICc values of 0.5 and 0.0 respectively” should maybe read “coinciding with the lowest AICc values gave d= 0.5 and 0.0 respectively”
Line 228: It should be “García-Dorado”
Line 285: It would be worthwhile mentioning that circular mating can have contributed to slow purging for survival, as it implies some equalization of family contributions and, therefore, some relaxation in natural selection besides increasing effective population size (see García-Dorado 2012 and Perez-Pereira et al. 2022 in Animal Conservation)
Line 383: Remove the hyphen in the link so that the link can work.
Author Response
Response to reviewer round2
The authors did a good job reviewing the manuscript that is now clearer and more convincing, and I think makes a remarkable contribution to the field. I think the manuscript can be published after a few minor issues are addressed (see below).
Authors are very pleased that the improvements of the manuscript are apparent and also that our contribution to the field is recognized by the reviewer.
Line 21: Replace with “reached the maximum possible value of 0.5 when estimated between 1992-1997”, as the method always estimates de initial parameter, although bias increase when too long periods are analysed. In the same line, in “expected in fitness”, remove “in”.
The suggested replacement is performed. In line 21 “in” has been removed.
Line 76: remove “that”.
“that” in line 76 has been removed.
Line 143: “coincided with the lowest AICc values of 0.5 and 0.0 respectively” should maybe read “coinciding with the lowest AICc values gave d= 0.5 and 0.0 respectively”
This change has been adapted.
Line 228: It should be “García-Dorado”
Authors apologise for this mistake. The name of the cited author is corrected.
Line 285: It would be worthwhile mentioning that circular mating can have contributed to slow purging for survival, as it implies some equalization of family contributions and, therefore, some relaxation in natural selection besides increasing effective population size (see García-Dorado 2012 and Perez-Pereira et al. 2022 in Animal Conservation)
In line 285 a short extension is added and the references are also extended.
Line 383: Remove the hyphen in the link so that the link can work.
Hyphen in line 383 has been removed.
Submission Date
11 October 2022
Date of this review
21 Dec 2022 12:04:37
Author would like to thank the reviewer yet again, the very constructive and positive evaluation!
Yours Faithfully,
István Nagy
22th Dec 2022.